# Characterizing Circulating microRNA Signatures of Type 2 Diabetes Subtypes

**DOI:** 10.3390/ijms26020637

**Published:** 2025-01-14

**Authors:** Fatima Sulaiman, Costerwell Khyriem, Stafny Dsouza, Fatima Abdul, Omer Alkhnbashi, Hanan Faraji, Muhammad Farooqi, Fatheya Al Awadi, Mohammed Hassanein, Fayha Ahmed, Mouza Alsharhan, Abdel Rahman Tawfik, Amar Hassan Khamis, Riad Bayoumi

**Affiliations:** 1College of Medicine, Mohammed Bin Rashid University of Medicine and Health Sciences, Dubai P.O. Box 505055, United Arab Emirates; fatima.sulaiman@students.mbru.ac.ae (F.S.); costerwell.khyriem@dubaihealth.ae (C.K.); stafny.dsouza@dubaihealth.ae (S.D.); fatima.abdul@dubaihealth.ae (F.A.); hanan.faraji@students.mbru.ac.ae (H.F.); 2Center for Applied and Translational Genomics, Mohammed Bin Rashid University of Medicine and Health Sciences, Dubai P.O. Box 505055, United Arab Emirates; omer.alkhnbashi@dubaihealth.ae; 3Dubai Diabetes Center, Dubai Health, Dubai P.O. Box 7272, United Arab Emirates; mhfarooqi@dubaihealth.ae; 4Endocrinology Department, Dubai Hospital, Dubai Health, Dubai P.O. Box 7272, United Arab Emirates; ffalawadi@dubaihealth.ae (F.A.A.); mmhassanein@dubaihealth.ae (M.H.); 5Pathology Department, Dubai Hospital, Dubai Health, Dubai P.O. Box 7272, United Arab Emirates; faaahmed@dubaihealth.ae (F.A.); maalsharhan@dubaihealth.ae (M.A.); 6Hamdan Bin Mohammed College of Dental Medicine, Mohammed Bin Rashid University of Medicine and Health Sciences, Dubai P.O. Box 505055, United Arab Emirates; abdel.tawfik@dubaihealth.ae (A.R.T.); amar.hassan@dubaihealth.ae (A.H.K.)

**Keywords:** Type 2 diabetes, circulating microRNA, T2D subtypes, pathophysiology, insulin resistance

## Abstract

Type 2 diabetes (T2D) is a heterogeneous disease influenced by both genetic and environmental factors. Recent studies suggest that T2D subtypes may exhibit distinct gene expression profiles. In this study, we aimed to identify T2D cluster-specific miRNA expression signatures for the previously reported five clinical subtypes that characterize the underlying pathophysiology of long-standing T2D: severe insulin-resistant diabetes (SIRD), severe insulin-deficient diabetes (SIDD), mild age-related diabetes (MARD), mild obesity-related diabetes (MOD), and mild early-onset diabetes (MEOD). We analyzed the circulating microRNAs (miRNAs) in 45 subjects representing the five T2D clusters and 7 non-T2D healthy controls by single-end small RNA sequencing. Bioinformatic analyses identified a total of 430 known circulating miRNAs and 13 previously unreported novel miRNAs. Of these, 71 were upregulated and 37 were downregulated in either controls or individual clusters. Each T2D subtype was associated with a specific dysregulated miRNA profile, distinct from that of healthy controls. Specifically, 3 upregulated miRNAs were unique to SIRD, 1 to MARD, 9 to MOD, and 18 to MEOD. Among the downregulated miRNAs, 11 were specific to SIRD, 9 to SIDD, 2 to MARD, and 1 to MEOD. Our study confirms the heterogeneity of T2D, represented by distinguishable subtypes both clinically and epigenetically and highlights the potential of miRNAs as markers for distinguishing the pathophysiology of T2D subtypes.

## 1. Introduction

Type 2 diabetes (T2D) is a chronic metabolic disorder characterized by persistent hyperglycemia, primarily due to insulin resistance and the failure of beta cells to secrete sufficient insulin [1]. Its global prevalence is rising, necessitating early diagnosis to prevent severe complications such as cardiovascular diseases, renal failure, retinopathy, and neuropathy [2]. According to the Global Burden of Disease (GBD) study in 2019, the global age-standardized prevalence rate of T2D was 5282.9 per 100,000 population [3]. The highest prevalence was recorded in the Middle East and North Africa (MENA) region (12.2%), projected to increase to 13.9% by 2045 [4]. In the United Arab Emirates (UAE), an overall prevalence of 16.3% has been reported, with the Northern Emirates showing rates as high as 25.1% among UAE nationals and 19.1% among expatriates [5]. T2D imposes a substantial economic burden on healthcare systems and negatively impacts quality of life, leading to physical limitations, emotional distress, and reduced social participation.

T2D is characterized by significant variability in individual phenotypes, stemming from a combination of genetic, environmental, and lifestyle influences [6,7,8,9]. The palette model of diabetes [10] posits that individuals develop T2D due to defects in multiple etiological pathways, including, but not limited to, beta cell function, beta cell mass, insulin action, glucagon secretion/action, incretin secretion/action, and fat distribution. The interplay of these defective pathways in varying degrees contributes to the diversity of phenotypes observed in individuals with T2D [9]. Recent efforts have aimed to deconstruct T2D heterogeneity into distinct clusters or subtypes [11,12,13]. Major studies have focused on phenotype and/or genotype approaches across various ethnicities [14,15,16,17,18,19]. In their first attempt to cluster T2D into subtypes within an Emirati cohort with long-standing T2D (~15 years), Bayoumi et al. [20] identified five clusters using five clinical parameters: fasting blood glucose (FBG), fasting serum insulin (FSI), body mass index (BMI), hemoglobin A1c (HbA1c), and age at diagnosis. The resulting clusters included severe insulin-resistant diabetes (SIRD), severe insulin-deficient diabetes (SIDD), mild age-related diabetes (MARD), mild obesity-related diabetes (MOD), and mild early-onset diabetes (MEOD). A significant overlap among clusters was reported in 57% of the cohort, while the remaining 43% exhibited cluster-specific membership.

To date, genetic variants identified by genome-wide association studies (GWAS) explain only approximately 15% of the total heritability of T2D.

Growing evidence suggests that epigenetic changes—such as DNA methylation, histone modifications, and small non-coding RNAs known as microRNAs (miRNAs)—are major contributors to many clinical features, pathogenesis, and heritability of T2D [21,22,23]. The study of miRNAs is particularly relevant because distinct metabolic patterns correspond to different miRNA expression profiles [21,24]. miRNAs are 18–22 nucleotide-long RNAs that regulate gene expression by binding to the 3′ untranslated region (UTR) of mRNA, thereby inhibiting translation. They have frequently been implicated in studies of T2D pathophysiology [25], playing fundamental roles in insulin biosynthesis and secretion [26], glucose homeostasis [27], and insulin resistance [28]. When compared with human plasma, serum, which is often described as a full-body biopsy, contains higher miRNA concentrations [29] and higher stability of circulating extracellular miRNAs even after multiple freeze–thaw cycles when archived for long-term storage [30]. Thus, the quantification of miRNAs in the serum of T2D patients enhances their potential as functional biomarkers for non-invasive diagnostic and prognostic applications, while offering an innovative solution to several unmet medical needs in diabetes care, such as insight into disease pathophysiology, identification of subtypes, personalization of treatment, improved disease monitoring and prognosis, and a potential for multi-dimensional diagnostics [31,32].

In this study, we examined the circulating miRNA profiles in T2D subtypes of Emirati patients previously described by Bayoumi et al. [20]. This is the first study of its kind investigating miRNA expression patterns in T2D patients in the Middle East. Our findings highlight the potential of circulating miRNAs as markers for the pathophysiology of T2D subtypes. Additionally, as miRNA expression is influenced by environmental factors, lifestyle, and disease states, this research provides valuable insights into the epigenetic mechanisms underlying T2D.

## 2. Results

The cohort characteristics of T2D patients of this study have been reported elsewhere [20]. Briefly, patients in the total T2D cohort tested (*n* = 348) had a mean duration of T2D of 14 years and had multiple comorbidities and complications, with every patient exhibiting at least 2 complications. Using unsupervised soft cluster analysis by Auto Cluster IBM Modeler in the SPSS software (version 18.0; IBM North America, New York, NY, USA), the 348 Emirati T2D patients were subtyped by Bayoumi et al. [20] by five etiological predictor variables: FBG, FSI, BMI, HbA1c, and age at diagnosis. Of the five clusters identified, the first four matched Ahlqvist et al. [14]’s subgroups (SIRD, SIDD, MARD, and MOD). A fifth new subtype, MEOD, was identified. An extensive overlap between these five clusters was observed. While 151/348 patients (43%) appeared in five distinct clusters, the remaining 197/348 patients (57%) showed varying degrees of overlap, with individuals appearing in two or more clusters. The clinical characteristics of the non-overlapping clusters are described in Appendix A. In this sub-cohort of non-overlapping subgroups (*n* = 151), 57% had hypertension, 58% had peripheral neuropathy, 35% had retinopathy, 14% had cardiovascular disease and 13% had chronic kidney disease. Most T2D patients tested were receiving multiple hypoglycemic drugs. The frequency at which these drugs were used in the sub-cohort of non-overlapping subgroups (*n* = 151) were as follows: metformin (89%), SGLT2 inhibitors (64%), DPP-4 inhibitors (44%), glipizide (40%), insulin (38%), GLP1-receptor agonists (30%), and thiazolidiedione (15%). A minimum of 66% of patients were receiving 3–4 drugs simultaneously. In addition, 87% were receiving lipid-lowering drugs and 56% were receiving antihypertensive treatment.

For the purpose of this study, nine patients at the peak of the distribution of each of the five non-overlapping phenotypic clusters (45/151) were selected. The primary clinical criterion identifying each subgroup is confirmed in the 45 selected patients (Table 1). The median age of diagnosis of T2D was oldest in the MARD subgroup (65 years) and youngest in MEOD (34 years). The MOD cluster had a mean BMI of 41.5 Kg/m^2^. SIDD showed the highest mean fasting blood glucose of 281 mg/dL, a mean HbA1c of 10.4% and the lowest mean β-cell function (HOMA-B) of 15. SIRD had the highest mean insulin resistance (HOMA-IR) of 15.1. All 45 T2D patients selected had two or more complications and undergoing treatment with 3–4 oral hypoglycemic drugs simultaneously.

### 2.1. Small RNA Expression Profile in Serum

Small RNA sequencing was performed on 45 T2D patients and 7 controls without diabetes, resulting in a small RNA profiling database of 686.7 million reads. Sample-wise read counts are reported in Appendix A. Two samples were excluded from further analysis due to poor RNA read quality. Quality control and filtering of raw reads yielded a total of 4,136,717 reads, of which 1,822,664 (44.04%) successfully aligned with a known small RNA species in the human reference genome (hg38) identifying various small RNAs such as piRNA, snRNA, snoRNA, miRNA, tRNAs and circular RNA. The overall small RNA expression profile indicated that certain classes were more highly expressed, with miRNAs being the most abundant. The proportion of miRNAs among total circulating small RNAs was approximately 35% in the SIRD cluster, 32% in SIDD, 35% in MARD, 35% in MOD, 35% in MEOD, and 50% in controls (Figure 1). A total of 1182 miRNAs were detected in the 50 samples. The miRNAs that had ≥10 copies (*n* = 430) were selected for downstream analysis. Of these, 253 miRNAs (58.83%) were expressed in all samples, 16 miRNAs in only controls and the remaining 161 in either one or multiple T2D clusters (Appendix A).

### 2.2. Differential miRNA Expression Patterns in T2D Subtypes and Controls

Out of the 430 circulating miRNAs expressed in serum, 57 were significantly differentially expressed between T2D patients and controls (*p* ≤ 0.05; log fold change ≥ 0.5). The distinct miRNA expression profiles are depicted in a heatmap (Figure 2) showing 17 upregulated and 40 downregulated miRNAs in T2D patients. The list of miRNAs is provided in Appendix A. The heatmap also highlights the heterogeneity in miRNA expression among T2D patients, contrasting with the relatively homogeneous profiles of non-T2D controls. This heterogeneity was further analyzed in patient phenotypic clusters described by Bayoumi et al. [20]

A total of 108 circulating miRNAs were significantly differentially expressed in different phenotypic clusters and controls (*p* ≤ 0.05; log fold change ≥ 0.5). Of those, 71 were upregulated and 37 were downregulated (Figure 3). Three miRNAs (hsa-miR-4660, hsa-miR-451a, hsa-miR-3146) were upregulated and 11 were downregulated in SIRD compared to other T2D clusters and controls (Table 2). While 12 of these were specific to SIRD, the downregulated hsa-miR-30e-3p also showed upregulation in MEOD and hsa-miR-1307-3p in controls. Nine miRNAs: hsa-miR-3143, hsa-miR-942-3p, hsa-miR-20b-5p, hsa-miR-576-5p, hsa-miR-548ay-5p, hsa-miR-548d-5p, hsa-miR-454-5p, hsa-miR-324-3p, hsa-miR-1843 were specifically downregulated in SIDD. Nine miRNAs: hsa-miR-548bc, hsa-miR-3614-5p, hsa-miR-6866-5p, hsa-miR-6741-5p, hsa-miR-320a-3p, hsa-miR-5000-3p, hsa-miR-320e, hsa-miR-576-3p were specifically upregulated in MOD. MARD differed from others by downregulation of hsa-miR-3928-3p, hsa-miR-378c and upregulation of hsa-miR-548s compared to all other groups. The highest number of differentially expressed miRNAs were observed in MEOD cluster (*n* = 19) with downregulation of hsa-miR-486-5p and upregulation of the remaining 18 miRNAs (Table 2). In healthy controls, 40 miRNAs were upregulated and 14 were downregulated compared to each of the T2D clusters. The differential expression patterns of all miRNAs were highly specific to their cluster and distinct from non-T2D controls.

Notably, many differentially expressed miRNAs were identified in the Human microRNA Disease Database (HMDD) V4.0 (http://www.cuilab.cn/hmdd; accessed on 17 April 2024). Several of these miRNAs have been previously linked to T2D, including hsa-miR-17 [33], hsa-miR-144 [34], hsa-miR-30e [35], hsa-miR-221 [36], hsa-miR-20b [37,38], hsa-miR-320a [39], hsa-miR-26b [40], hsa-miR-181a [41], hsa-miR-191 [42], hsa-miR-199a [43], hsa-miR-454 [44], hsa-miR-150 [45], hsa-miR-143 [46], hsa-miR-223 [47], hsa-miR-29a [48] and hsa-miR-338 (Appendix A). Additionally, other identified miRNAs, such as hsa-miR-214, hsa-miR-129, hsa-miR-628, hsa-miR-615, hsa-miR-885 and hsa-miR-24, were associated with nephropathy, retinopathy, ischemic heart disease, fatty liver disease and diabetic ulcers, respectively. However, some differentially expressed miRNAs reported in this study, such as hsa-miR-6852, hsa-miR-3146, hsa-miR-3143, hsa-miR-548ay, hsa-miR-548s, hsa-miR-3928, hsa-miR-3614, hsa-miR-6866, hsa-miR-6837, hsa-miR-1197, hsa-miR-3177, hsa-miR-4685, hsa-miR-6741, and hsa-miR-6786, have not been previously linked to any disease phenotype/pathology (Appendix A).

### 2.3. miRNA-Target Genes and Pathway Analysis

Potential target genes of the differentially expressed miRNA were identified using miRNET and MIENTURNET tools. By combining the results from both tools, we identified a total of 868 miRNA-target genes. Pathway analysis and ORA of these miRNA-target genes identified several biologically significant pathways. For further assessment, all pathways related to cancer and/or carcinogenesis were excluded. The analysis revealed enrichment of T2D-associated pathways including PI3K-AKT, MAPK, mTOR, HIF-1, AGE-RAGE, WNT and Hippo signaling (Figure 4). Subsequently, the relationship between differentially expressed miRNAs and their target genes that overlap with the T2D-related pathways was visualized via a heatmap (Appendix A).

### 2.4. Tissue Distribution of Circulating miRNA

To determine the tissue of origin for the differentially expressed circulating miRNAs, the miRNA tissue expression dataset from the TissueAtlas study was used. The TissueAtlas dataset was filtered to match the list of differentially expressed miRNA focusing on peripheral tissues such as liver, pancreas, muscle and adipose tissue. No specific miRNA showed significant enrichment in any organ. However, hsa-miR-1468-5p and hsa-miR-576-5p were highly expressed in the liver. In addition, hsa-miR-1271-5p was highly expressed in muscle, and finally, hsa-miR-129-5p was found to be expressed in the pancreas (Appendix A). Further evaluations are needed to confirm the tissue specificity of these miRNAs.

### 2.5. Novel miRNA Detected

A total of 13 novel miRNAs were identified across all clusters. These miRNAs had reads aligned to the mature, star and loop regions, exhibited a significant RANDFOLD *p*-value, showed no sequence homology with rRNA or tRNA and had a minimum read count of 10. The mature sequences of these 13 miRNAs displayed considerable overlap (Figure 5). The pre-miRNA and star sequences are shown in Appendix A.

## 3. Discussion

In this study, we examined the circulating miRNA expression profiles of five previously reported subtypes of long-standing T2D patients—SIDD, SIRD, MARD, MOD and MEOD [20] compared to controls with no family history of the disease, using high-throughput small RNA sequencing. This approach contrasts with many studies that focus on miRNA expression profiles in non-subtyped T2D patients [49]. Our results revealed that each T2D subtype is associated with a specific miRNA profile distinct from that of healthy controls. In total, 108 circulating miRNAs were significantly differentially expressed across the different T2D subtypes and controls. Specifically, 3 upregulated miRNAs were unique to SIRD, 1 to MARD, 9 to MOD and 18 to MEOD. Among the downregulated miRNAs, 11 were specific to SIRD, 9 to SIDD, 2 to MARD and 1 to MEOD. In healthy controls, 40 miRNAs were upregulated, and 14 were downregulated compared to all T2D clusters. Beyond diabetes-related pathways, most of the identified dysregulated miRNAs play significant roles in regulating pathways such as PI3K/AKT, Wnt/β-catenin, MAPK/ERK, JAK/STAT, p53, NF-κB, notch signaling and TGF-β signaling, thereby influencing processes like cell proliferation, migration, apoptosis and immune response.

As seen in many previous studies, our findings confirm that miRNA dysregulation plays a critical role in T2D pathogenesis. Of the 57 differentially expressed miRNAs, 40 (~70%) were downregulated in T2D patients compared to non-T2D controls, consistent with Sala et al. who reported that most differentially expressed miRNAs in diabetes are downregulated, likely due to DNA methylation mechanisms affecting miRNA loci [21]. Some of the downregulated miRNAs identified in this study, such as hsa-miR-615-3p, hsa-miR-335-3p, hsa-miR-130b-5p, hsa-miR-17-3p, hsa-miR-937, hsa-miR-1307-3p, hsa-miR-328-3p, hsa-mir-139-3p, hsa-miR-548ag and hsa-miR-548p have been previously linked to T2D mechanisms, including glucose metabolism, insulin signaling and insulin resistance [50]. Additionally, they are associated with complications like diabetic retinopathy, nephropathy, foot ulcers, and ischemic heart disease [50,51,52,53,54,55,56]. For instance, the downregulation of hsa-miR-130b-5p in morbidly obese patients was significantly associated with elevated serum cholesterol levels in a Qatari population [57]. Furthermore, a 5-year prospective study found that hsa-miR-1307-3p could predict T2D onset years before its manifestation [58]. While some miRNAs like hsa-miR-937, hsa-miR-615-3p and hsa-miR-548ag were downregulated in this study, they have been reported as upregulated in others, potentially contributing to metabolic disorders, ischemic heart disease, and obesity-associated T2D [59,60]. In contrast, the upregulated miRNAs in our T2D cohort, such as hsa-miR-150-3p, hsa-miR-125a-3p, hsa-miR-223-5p, hsa-miR-143-3p, hsa-miR-29a-3p and hsa-miR-24-3p, are known to be involved in insulin secretion/signaling, glucose homeostasis, insulin resistance, β-cell dysfunction and lipid metabolism [24,34,61,62,63,64,65].

T2D subtype-specific miRNA expression patterns revealed unique signatures. For example, the SIRD cluster had three upregulated and 11 downregulated miRNAs, including hsa-miR-224, hsa-miR-766, and hsa-miR-1307, which have been previously associated with T2D [66,67,68]. However, hsa-miR-6852 and hsa-miR-3146, identified in Emirati patients, are novel findings. The upregulation of miRNA-451a in T2D patients was consistent with prior studies which showed that it was also elevated in patients with non-alcoholic fatty liver disease, increasing their risk for T2D [69,70]. Moreover, hsa-miR-30e-3p, downregulated in the SIRD subtype, has been consistently downregulated in several human profiling studies and linked to prediabetes [24,35]. In contrast to our findings, miR-221 has been reported as upregulated in both T2D and metabolic syndrome patients [36,71]. The enrichment of glycolysis and gluconeogenesis pathways in the SIRD cluster suggests that dysregulation of these pathways play a central role in the development of insulin resistance which is the defining feature of this subtype.

The SIDD cluster showed the downregulation of nine miRNAs, which were distinct from other clusters. Consistent with our result, He et al. [37] have reported hsa-miR-20b-5p to be downregulated in T2D. It targets genes in the PI3K-AKT-MAPK pathways, contributing to insulin-stimulated glucose metabolism [38]. Interestingly, many of the circulating miRNAs specific to the SIDD cluster have been previously associated with psychological conditions such as anxiety, depression, and schizophrenia [72].

The MARD cluster was characterized by one upregulated (hsa-miR-548s) and two downregulated (hsa-miR-3928-3p and hsa-miR-378c) miRNAs. In previous studies, hsa-miR-378c has been shown to be downregulated in T2D [37], regulate adiponectin expression, and play a role in T1D pathogenesis [73,74].

The MOD cluster exhibited nine upregulated miRNAs. Previous research has shown that hsa-miR-320a-3p, which was upregulated in MOD cluster is also elevated in T2D patients and negatively correlates with BMI, FBG and HOMA-IR in obese individuals [39]. This cluster also showed an enrichment of the TNF-α pathway, which aligns with the role of inflammatory cytokines in obesity-related insulin resistance [75].

The MEOD cluster had the highest number (*n* = 19) of differentially expressed miRNAs, with only one downregulated miRNA (hsa-miR-486-5p) known to promote pancreatic cell proliferation, increase insulin sensitivity and inhibit apoptosis [76] and which was also reported as downregulated in T2D by He et al. [37]. The has-miR-144-5p upregulated in MEOD has also been shown to be upregulated in T2D patients and inhibit the expression of insulin receptor substrate 1 [50]. Other upregulated miRNAs like hsa-miR-26b-5p, hsa-miR-570-3p, hsa-miR-191-5p and hsa-miR-181a-5p were found to be involved in insulin signaling and resistance, glycemic impairment and significant risk for T2D [24,34,40,41,42,77].

A notable observation, in this study, was the absence of commonly reported circulating miRNAs such as hsa-miR-375 and hsa-miR-126. Furthermore, we identified 13 novel miRNAs with significant RNAfold *p*-values using miRDeep2, although these were not common across all clusters.

T2D is a heterogeneous disease with multiple subtypes, each with distinct overlapping etiologies. Individual patients within the same subtype can display unique miRNA expression patterns, highlighting the complexity of the disease and the need for personalized approaches to treatment. However, this study has its own limitations. Although hemolysis of serum samples was negated visually, a more sophisticated method like spectrometry can ensure no hemolysis. We also cannot rule out the potential release of platelet derived miRNAs in the serum samples since RNA released during the coagulation process may change the true repertoire of circulating miRNA [29]. The results are restricted to a small subset of samples that represent each T2D subtype warranting a validation of the expression patterns in a larger sample size by RT-qPCR.

Despite the small sample size, this study demonstrates the differences in miRNA expression patterns between T2D subtypes and non-T2D controls. Unlike many studies that use targeted techniques like qRT-PCR, we employed small RNA sequencing to evaluate both known and novel miRNAs, offering a more comprehensive view. This study provides a foundation for future research into the miRNA profiles of T2D subtypes, especially in newly diagnosed T2D patients emphasizing the need for further investigation into the specific roles of these miRNAs in disease pathogenesis.

## 4. Materials and Methods

### 4.1. Study Design

The participants of this study were recruited during their clinical visits to Dubai Diabetes Center (DDC) and Dubai Hospital (DH), Dubai, United Arab Emirates between January 2020 and December 2022 as described in our previous study [20]. The participants consisted of only Emirati nationals diagnosed with T2D per the American Diabetes Association criteria [78]. All participants were provided with detailed explanations prior to obtaining written informed consent. The clinical data of the patients were obtained by a healthcare practitioner through face-to-face questionnaires and supplemented by demographic information, medical history and laboratory investigations. The study was approved by Dubai Scientific Research Ethics Committee (DSREC) and the Mohammed Bin Rashid University of Medicine & Health Sciences’ Institutional Review Board (MBRU-IRB).

### 4.2. Subjects’ Selection

All patients included in the study tested negative for GAD antibodies (ELISA Test Kit; Demeditec Diagnostics, GmbH, Kiel, Germany) and were diagnosed with T2D (fasting blood glucose (FBG) ≥ 125 mgdL, Hemoglobin A1c (HbA1c) ≥ 6.5%) by a registered medical practitioner. Patients with conditions causing secondary diabetes and other types of diabetes (MODY, LADA) were excluded. The patients exhibited at least two co-morbidities (hypertension, hyperlipidemia, obesity) or complications (micro- and macrovascular) and were under treatment by different medications (metformin, thiazolidines, SGLT2 inhibitors and GLP-1 agonists).

Patients: The T2D subtypes in this study cohort have been previously reported by Bayoumi et al. [20]. Briefly, five phenotypic subtypes were predicted in 348 Emirati patients with long-standing T2D by applying the Auto Cluster model in IBM SPSS Modeler software (version 18.0; IBM North America, New York, NY, USA). The five predictor variables (fasting blood glucose (FBG), fasting serum insulin (FSI), body mass index (BMI), HbA1c and age at diagnosis) used for clustering assigned 151 patients to non-overlapping clusters: SIRD (*n* = 25), SIDD (*n* = 40), MARD (*n* = 24), MOD (*n* = 23), and MEOD (*n* = 39) (Figure 1). For the purpose of this study, nine patients at the peak of the distribution of each of the five phenotypic clusters were selected. In addition, 7 healthy adults with no diabetes and no history of T2D in their nuclear family, including their four grandparents, as confirmed by questionnaire, were recruited as controls.

Controls: Research suggests that family history is a strong independent predictor of T2D, and while there are many studies that examined this risk factor, few studies have identified true diabetic controls that have no family history of diabetes. To identify individuals without a family history of T2D, a brief survey was conducted among individuals that belong to large extended families. Data were obtained confirming no history of diabetes in the individual, his/her parents, siblings, four grandparents, maternal and paternal aunts, and uncles. Our results (unpublished) confirmed results of a previous study [79] where only 15% of participants had no family history of T2D and, therefore, were selected as controls.

### 4.3. Small RNA Sequencing

Blood was collected from study participants in Vacutainers without preservative. Serum was separated by centrifugation (2000× *g* for 10 min at 4 °C) from whole clotted blood (~4 mL) and stored in cryovials at −80 °C. Cell-free circulating small RNAs were extracted from 200 µL of serum using Plasma/Serum RNA Purification Kit (Norgen Biotek, Thorold, ON, Canada) following the manufacturer’s protocol and quality-checked by Bioanalyzer (Agilent Technologies, Santa Clara, CA, USA). The cDNA library preparation and sequencing were performed by Norgen Biotek, Canada as follows: (i) Libraries were created using the Small RNA Library Prep Kit for Illumina (Norgen Biotek, Canada) according to the manufacturer’s protocol. (ii) The concentration of libraries was measured using Bioanalyzer. (iii) Single-end small RNA sequencing was performed using High Output Kit v2 (51 Cycles using a 75-Cycle Kit) on NextSeq 500/550 (Illumina Inc., San Diego, CA, USA).

### 4.4. Bioinformatic Analysis

The following steps were followed for analysis of the small RNA sequence data.

#### 4.4.1. Pre-Processing of Data

Raw reads were received in a gzip compressed fastq format from Norgen Biotek Corp. (Thorold, ON, Canada). Raw reads were quality-checked with FastQC V0.12.0 (https://www.bioinformatics.babraham.ac.uk/projects/fastqc/; accessed on 27 March 2024). Low-quality reads and adapter sequences were removed by the *bbduk* command from BBmap V39.06 (https://sourceforge.net/projects/bbmap/; accessed on 27 March 2024). The adapter sequences used for the trimming are provided in the adapter.fa file which includes all the Illumina indexes. The options used for the trimming were right trimming (3′ end), and k-mer was set to 15, i.e., the contaminate’ sequence should at least have a length of 15 bp, with exception to sequences at the end of the reads where k-mer was set at 10, hamming distance (mismatch) of one and quality score of 30. The reads were further ‘de-duplicated’ using the *dedupe* command from BBmap tool. The de-duplicated fastq files were then used for alignment.

#### 4.4.2. Read Alignment

Alignment of the reads to miRNA sequences was carried out using miRDeep2 V2.0.0.4 [80]. The high-quality deduplicated reads were aligned to the human reference genome (hg38; last accessed 19 January 2024) with the *mapper.pl* function of miRDeep2 [81]. The aligned reads were then mapped to mature miRNA sequences retrieved from miRbase V22.1 (last accessed 18 March 2024) [82] with mirDeep2.pl to generate read counts. The raw read counts were normalized as reads per kilobase million (RPKM). To confirm that the overall read distribution of the miRNAs across all samples followed normal distribution after RPKM normalization (reads per kilobase of transcript per million reads mapped), we calculated the relative log expression (RLE) values of each miRNA, which were then plotted using a boxplot for each sample with the *plotRLE* function from the EDASeq v2.36.0 [83]. The values from the above function were then passed to ggplot2 v3.5.1 [84] for making the boxplot to highlight sample and phenotypic cluster on the plot.

#### 4.4.3. Small RNA Biotypes Identification in Serum

Alignment of the deduplicated reads to different small RNA biotypes was performed using the exceRpt pipeline [85]. Reference genome version of hg38 was used along with the precompiled small RNA library available on the exceRpt website. The total read count for miRNA, tRNA, piRNA, circularRNA, misc_RNA, snRNA, snoRNA, scaRNA, scRNA, protein_coding, and rRNA were retrieved using an in-house R package. The stacked bar graph was created using ggplot2 v3.5.1. 

#### 4.4.4. Differential Expression of miRNAs

The differential expression of miRNA was estimated by SEURAT V4.4.0 [86,87] in two phases: (1) all T2D patients versus non-T2D controls; (2) between patients in five phenotypic clusters and non-T2D control. To better understand and control the diversity of the miRNAs amongst the subjects at the level of their association to each of the phenotypic clusters as well as at overall T2D condition or lack thereof, we adopted a method which is commonly used in single-cell sequencing analysis. In this method, we calculated the average expression of miRNAs either within a given phenotype or condition which is expressed in at least 60% of the samples and with a minimum of 10 reads aligned. Then, we statistically compared the differences in the average expression between the phenotypes or conditions.

In the case of all T2D patients versus control, all the expression values of miRNA expressed in 60% of the samples within T2D were compared with the controls, whereas between clusters versus control, the same percentage frequency cutoff of the miRNA for each phenotypic cluster and the control were compared against each other. This provided an overview of the divergent miRNA profile between the T2D condition as well as each phenotypic cluster against the control. The *FindAllMarkers* function in the SEURAT package was used to perform the differential expression analysis complimented with the *t*-test. The top differentially expressed miRNAs for each cluster were fetched giving weight to the log fold change (logFC). A heatmap was plotted using the scaled expression to visualize the differential expression between clusters using *Doheatmap* function from SEURAT package.

#### 4.4.5. Identification of miRNA-Target Genes and Pathway Analysis

Target genes of differentially expressed miRNAs were identified by miRNet v2.0 [88] and MIENTURNET [89]. For miRNet, a confidence score threshold of 6 was applied to signify the strength of miRNA-to-target genes association, and a *p*-value cutoff of 0.05 was applied for MIENTURNET. Pathway analysis on target genes was done using webgestalt [90]. Over-representation analysis (ORA) was carried out with the KEGG database [91]. BH correction on the *p*-values and 0.05 was used as the cutoff.

#### 4.4.6. Tissue Atlas of the Upregulated and Downregulated miRNA Between Clusters

To explore the expression of the differentially regulated miRNA in our datasets in the peripheral tissues such as liver, pancreas, muscle and adipocyte, we leveraged the expression dataset compiled in the TissueAtlas. The TissueAtlas dataset was filtered out to match with our list of differentially expressed miRNAs. miRNAs with RPKM less than 10 in the TissueAtlas were removed and a boxplot was generated to visualize the expression patterns.

#### 4.4.7. Novel miRNA Detection

The unaligned reads to human reference genome were used for predicting the previously unreported novel miRNAs in this dataset using miRDeep2 with default parameters. Only miRNAs that had reads aligned to the mature, star and loop regions exhibited a significant RNAfold *p*-value, lacked sequence homology with rRNA or tRNA, and had at least 10 read counts were considered novel. Thirteen miRNAs met the above criteria. Multiple sequence alignment of the mature and pre-mature sequences of the 13 selected miRNA was carried out using the R package msa V1.36.1 [92], and the resulting alignment was passed to ggmsa V1.3.4 [93] for visualization.

### 4.5. Statistical Analysis

The *t*-test was used to calculate the significance of the differentially expressed miRNAs. Subsequence pathway analysis and novel miRNA detection statistics were carried out within the tools that were used for those specific tasks.

## 5. Conclusions

This study identifies the dysregulation of 108 circulating microRNAs across various subtypes of type 2 diabetes (T2D). Notably, 36 miRNAs are linked to T2D for the first time, including several whose molecular functions remain poorly understood in the existing literature. The variation in circulating miRNA expression among T2D subtypes highlights the disease’s heterogeneity. These differentially expressed miRNAs created distinct signatures that differentiated each of the five T2D subtypes from a homogeneous control group. The findings underscore the potential of circulating miRNAs as biomarkers for distinguishing the pathophysiology of T2D subtypes. Furthermore, since miRNA expression is influenced by environmental factors, lifestyle choices, and disease states, this research provides valuable insights into the epigenetic mechanisms that underlie T2D. Pathway analysis confirmed the involvement of these miRNAs in insulin resistance, inflammation, and obesity-related pathways. Additionally, some of the identified miRNAs are associated with T2D complications such as nephropathy, retinopathy, and cardiovascular diseases, which could enhance our ability to predict and manage these complications more effectively. Moreover, this study opens promising avenues for developing targeted miRNA therapeutics for each subtype, by modulating the levels of the identified miRNAs. Such interventions could potentially restore normal metabolic functions and improve insulin sensitivity.

## Figures and Tables

**Figure 1 ijms-26-00637-f001:**
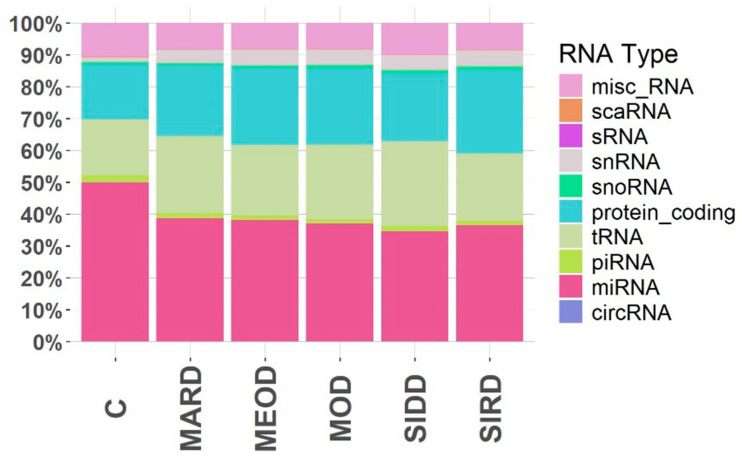
Cluster-wise small RNA read distribution in serum of type 2 diabetes patients and controls. The percentage distribution of reads aligning to small RNA biotype is shown along the Y-axis as stacked bars for each phenotypic cluster. The X-axis shows clusters such as mild early-onset diabetes (MEOD), mild age-related diabetes (MARD), mild obesity-related diabetes (MOD), severe insulin-resistant diabetes (SIRD), severe insulin-deficient diabetes (SIDD) and control (C).

**Figure 2 ijms-26-00637-f002:**
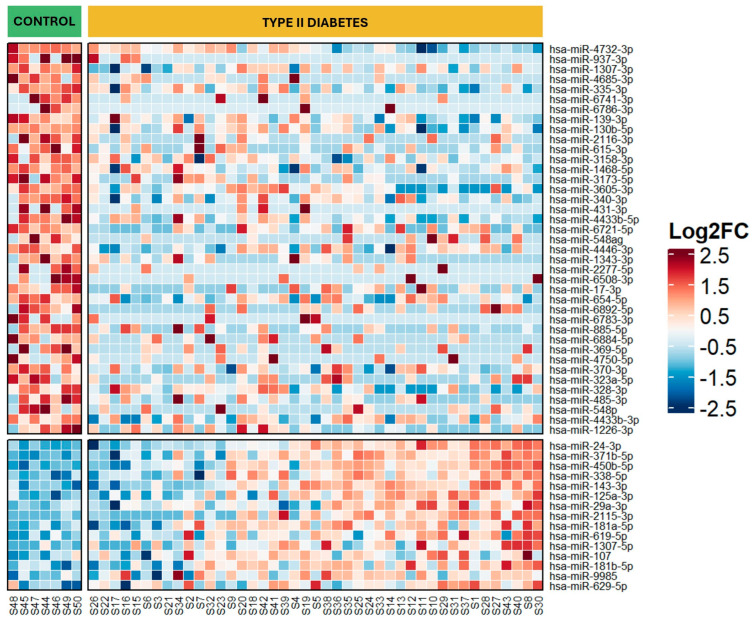
Differential expression profile of circulating miRNAs in type 2 diabetes versus control. The log2Fold change (log2FC) of each differentially expressed miRNA between the Control and Type 2 diabetes (T2D) groups is shown as heatmap where the red indicates upregulated miRNAs, blue represents downregulated miRNAs and white represents no expression change. The two vertical groups correspond to control (green bar) and T2D (Yellow bar). The individual vertical column in each group indicates the results of a single patient or subject and each horizontal row represents a single miRNA.

**Figure 3 ijms-26-00637-f003:**
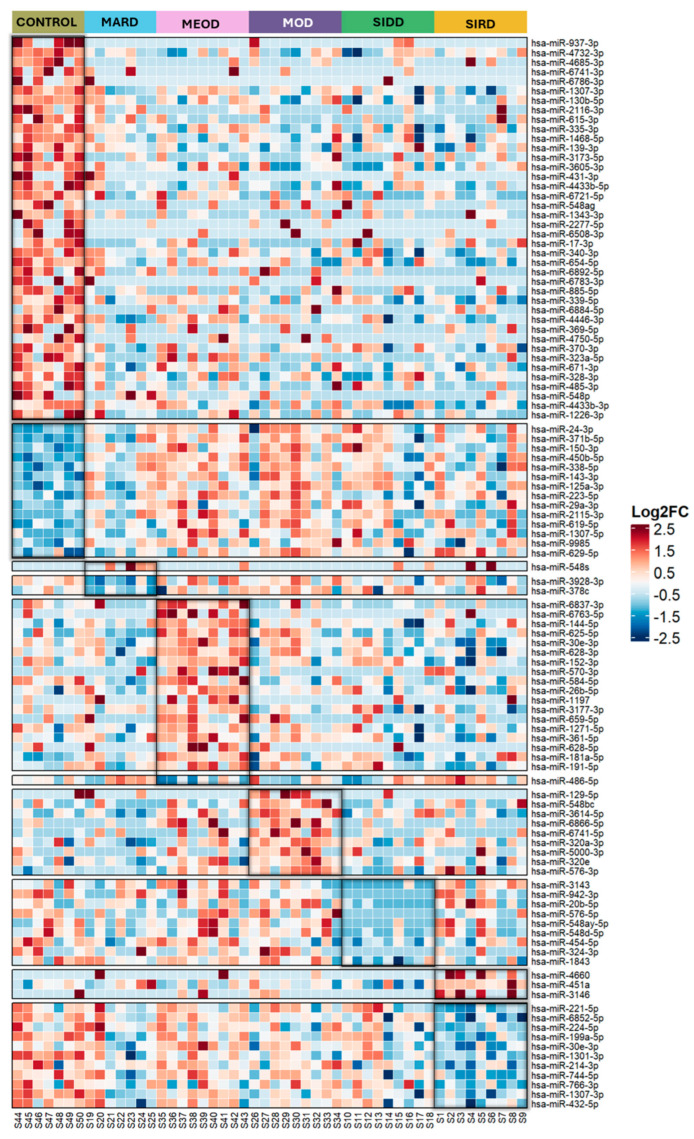
Type 2 diabetes cluster-wise differential expression profile of circulating miRNAs. The log2Fold change (log2FC) of each differentially expressed miRNA between the five type 2 diabetes (T2D) clusters and control is shown as heatmap where the red indicates upregulated miRNAs, blue represents downregulated miRNAs and white represents no expression change. Cluster-specific differentially expressed miRNA is highlighted in black. Each column indicates the results of a single patient or subject.

**Figure 4 ijms-26-00637-f004:**
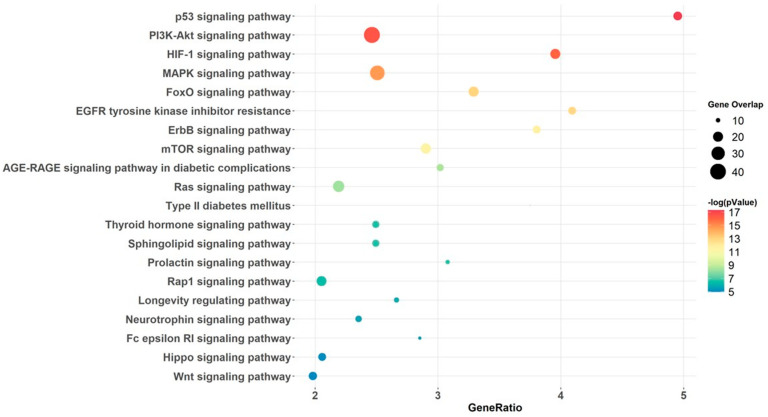
miRNA-pathway analysis for five type 2 diabetes clusters and control. The size of dots indicates the number of target genes overlapping with the gene list for each pathway. The color of the dots indicates the range of *p*-values, with green being closer to 0.05 and red being farther away. The Y-axis lists the different pathways, and the X-axis shows the ratio of genes shared between the target gene list and the specific pathway.

**Figure 5 ijms-26-00637-f005:**
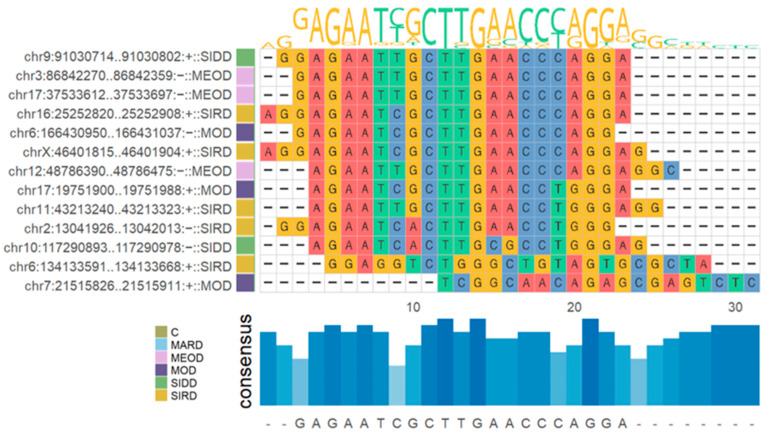
Multiple sequence alignment of the predicted novel miRNA. The consensus between the mature sequences of the predicted 13 significant novel miRNA is shown.

**Table 1 ijms-26-00637-t001:** Clinical characteristics of T2D patients involved in the study, classified by unsupervised clustering approach in IBM SPSS Modeler.

	SIRD	SIDD	MARD	MOD	MEOD
Number of patients, *n*	9	9	9	9	9
Age at diagnosis (years)	47.2 (12.6)	41.1 (10.2)	**63.4** (8.6)	36.0 (12.9)	**32.2** (6.6)
Body mass index (Kg/m^2^)	33.2 (4.1)	31.3 (6.4)	28.6 (3.1)	**42.6** (5.3)	23.9 (1.6)
Fasting blood glucose (mg/dL)	163 (40)	**287** (68)	124 (24)	126 (36)	109 (26)
HbA1c (%)	7.1 (1.1)	**10.7** (2.3)	6.5 (0.7)	7.0 (0.6)	6.2 (0.6)
Fasting insulin (mIU/mL)	**44.4** (9.0)	9.1 (5.1)	12.5 (5.7)	12.0 (7.1)	7.4 (2.6)
HOMA_IR	**17.9** (6.7)	6.8 (4.8)	4.1 (2.5)	3.9 (2.8)	1.9 (0.7)
HOMA_B	198.0 (147.2)	**14.0** (6.7)	75.7 (31.7)	91.3 (97.9)	76.4 (59.1)

SIRD, severe insulin-resistant diabetes; SIDD, severe insulin-deficient diabetes; MARD, mild age-related diabetes; MOD, mild obesity-related diabetes; MEOD, mild early-onset diabetes; HOMA-B, homeostasis model assessment of beta-cell function; HOMA-IR, homeostasis model assessment of insulin resistance. All values are shown as mean (±standard deviation). The parameter defining the cluster is highlighted in bold.

**Table 2 ijms-26-00637-t002:** List of differentially expressed circulating miRNA in control and T2D clusters.

	Upregulated	Downregulated
Control	hsa-miR-937-3p, hsa-miR-6786-3p, hsa-miR-615-3p, hsa-miR-17-3p, hsa-miR-370-3p, hsa-miR-6741-3p, hsa-miR-369-5p, hsa-miR-1343-3p, hsa-miR-6721-5p, hsa-miR-1468-5p, hsa-miR-431-3p, hsa-miR-6783-3p, hsa-miR-485-3p, hsa-miR-335-3p, hsa-miR-130b-5p, hsa-miR-4685-3p, hsa-miR-2277-5p, hsa-miR-6892-5p, hsa-miR-654-5p, hsa-miR-139-3p, hsa-miR-6508-3p, hsa-miR-548ag, hsa-miR-1226-3p, hsa-miR-3605-3p, hsa-miR-4446-3p, hsa-miR-2116-3p, hsa-miR-548p, hsa-miR-323a-5p, hsa-miR-340-3p, hsa-miR-4433b-3p, hsa-miR-3173-5p, hsa-miR-4750-5p, hsa-miR-885-5p, hsa-miR-339-5p, hsa-miR-1307-3p, hsa-miR-4433b-5p, hsa-miR-6884-5p, hsa-miR-4732-3p, hsa-miR-328-3p, hsa-miR-671-3p	hsa-miR-2115-3p, hsa-miR-619-5p, hsa-miR-450b-5p, hsa-miR-150-3p, hsa-miR-338-5, hsa-miR-371b-5p, hsa-miR-1307-5p, hsa-miR-125a-3p, hsa-miR-223-5p, hsa-miR-143-3p, hsa-miR-29a-3p, hsa-miR-9985, hsa-miR-629-5p, hsa-miR-24-3p
MOED	hsa-miR-6763-5p, hsa-miR-6837-3p, hsa-miR-625-5p, hsa-miR-1271-5p, hsa-miR-628-3p, hsa-miR-570-3p, hsa-miR-1197, hsa-miR-144-5p, hsa-miR-30e-3p, hsa-miR-26b-5p, hsa-miR-628-5p, hsa-miR-659-5p, hsa-miR-3177-3p, hsa-miR-152-3p, hsa-miR-361-5p, hsa-miR-584-5p, hsa-miR-181a-5p, hsa-miR-191-5p	hsa-miR-486-5p
MARD	hsa-miR-548s	hsa-miR-3928-3p, hsa-miR-378c
MOD	hsa-miR-548bc, hsa-miR-3614-5p, hsa-miR-6866-5p, hsa-miR-6741-5p, hsa-miR-320a-3p, hsa-miR-5000-3p, hsa-miR-320e, hsa-miR-576-3p	-
SIRD	hsa-miR-4660, hsa-miR-451a, hsa-miR-3146	hsa-miR-221-5p, hsa-miR-6852-5p, hsa-miR-224-5p, hsa-miR-199a-5p, hsa-miR-30e-3p, hsa-miR-1301-3p, hsa-miR-214-3p, hsa-miR-744-5p, hsa-miR-766-3p, hsa-miR-1307-3p, hsa-miR-432-5p
SIDD	-	hsa-miR-3143, hsa-miR-942-3p, hsa-miR-20b-5p, hsa-miR-576-5p, hsa-miR-548ay-5p, hsa-miR-548d-5p, hsa-miR-454-5p, hsa-miR-324-3p, hsa-miR-1843

MEOD, mild early-onset diabetes; MARD, mild age-related diabetes; MOD, mild obesity-related diabetes; SIRD, severe insulin-resistant diabetes; SIDD, severe insulin-deficient diabetes.

## Data Availability

The raw data supporting the conclusions of this article will be made available by the authors on request.

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
