# Peer review of "Characterizing Circulating microRNA Signatures of Type 2 Diabetes Subtypes"

_ijms, 2025, doi:10.3390/ijms26020637_

Round 1

Reviewer 1 Report

Comments and Suggestions for Authors

Characterizing circulating microRNA signatures of type 2 diabetes subtypes.

The objetive of this study was identify T"D cluster specific miRNA expresson signatures for the five clinical subtypes that characterize the underlying pathophysiology of long standing T2D:  severe insulin-resistant diabetes (SIRD), severe insulin deficient diabetes (SIDD), mild age related diabetes (MARD), mild obetity related diabetes (MOD), and mild early onset diabetes (MEOD). They analized the circulating microRNAs in 9 subjects of each T2D cluster and 7 healthy controls by single-end small RNA sequencing. Their results showed that each type of diabetes had a specific miRNA profile. And they concluded that T2D subytpes have clinically and epigenetically diferences and highlights the potential of miRNAs as markers for distinguishing the pathophysiology of T2D subtypes.

Author Response

Rviewers Comment; The objective of this study was identifying T2D cluster specific miRNA expression signatures for the five clinical subtypes that characterize the underlying pathophysiology of long standing T2D: severe insulin-resistant diabetes (SIRD), severe insulin deficient diabetes (SIDD), mild age-related diabetes (MARD), mild obesity related diabetes (MOD), and mild early onset diabetes (MEOD). They analyzed the circulating microRNAs in 9 subjects of each T2D cluster and 7 healthy controls by single-end small RNA sequencing. Their results showed that each type of diabetes had a specific miRNA profile. And they concluded that T2D subtypes have clinically and epigenetically differences and highlights the potential of miRNAs as markers for distinguishing the pathophysiology of T2D subtypes.

Response: We thank the reviewer for taking the time to review our manuscript. We truly appreciate his thorough and thoughtful evaluation, and we are pleased to know that he found no concerns or clarifications needed.

Reviewer 2 Report

Comments and Suggestions for Authors

The study investigates circulating microRNA (miRNA) profiles in patients with Type 2 Diabetes (T2D), divided into five clinical subtypes: SIRD (Severe Insulin-Resistant Diabetes), SIDD (Severe Insulin-Deficient Diabetes), MARD (Mild Age-Related Diabetes), MOD (Mild Obesity-Related Diabetes), and MEOD (Mild Early-Onset Diabetes). Using small RNA sequencing of serum samples from 45 patients and 7 healthy controls, the research identified: 108 differentially expressed miRNAs, with 71 upregulated and 37 downregulated; unique miRNA profiles for each T2D subtype, distinct from those of healthy controls. The discovery of 13 novel miRNAs not previously reported is also included. In addition, authors have performed enrichment analyses to identify metabolic pathways associated with T2D, such as PI3K-AKT, MAPK, and mTOR.

Although the study is focusing on a topic of tremendous clinical significance, several issues need to be solved before the paper may be published:

1.     Authors should explain in more details what is the advantage of discriminating those T2D patients they are analysing; in other words, which is the unmet medical need that miRNA quantification can help to solve.

2.     T2D subjects need to be better described for confounding factors, such as treatments and presence of T2D complications.

3.     Serum is not optimal for microRNA quantification, since it contains all the molecules released upon platelet activation. Are authors able to quantify “plasma” circulating microRNAs in at least a subset of samples, to make sure they can obtain similar data? Furthermore, the haemolysis issue is also crucial here, since miR-451 is among the differentially expressed miRNAs, but it is also known to be highly expressed in haemolyzed plasma/serum samples.

4.     Patients’ description. Table 1 is not clear enough: explain the classification based on the unsupervised clustering approach. The reference 20 is too relevant for this present work for just being cited. Please explain the bases for your present effort.

5.     The sentence “On applying a read count cut-off of ≥10 per sample the total number of identified miRNAs reduced from 1182 to 430” is unclear. Please explain better. It would be relevant to evaluate how many miRNAs are co-expressed in all samples.

6.     The legend of the heatmap of Figure 3 is misleading. It is not clear if T2D and Controls are subdivided in vertical (by colour) or in horizontal (by names on the side). Please clarify.

7.     A validation phase (also by a different platform of quantification, such as RT-qPCR) may significantly strengthen the study. Moreover, authors may give a hint on the actual number of miRNA molecule per serum volume unit. This is relevant, since a biomarker should also be robustly expressed and easily detectable.

8.     Discussion is vague. Which of the miRNAs here identified are also found dysregulated (and with the same direction) in other studies?

Author Response

  1. Authors should explain in more details what is the advantage of discriminating those T2D patients they are analysing; in other words, which is the unmet medical need that miRNA quantification can help to solve.

Response: The reviewer’s suggestion is accepted and has been addressed in the Introduction section as follows.

Lines 79- 85: “Thus, the quantification of miRNAs in the serum of T2D patients enhances their potential as functional biomarkers for non-invasive diagnostic and prognostic applications, while offering an innovative solution to several unmet medical needs in diabetes care, such as insight into disease pathophysiology, identification of subtypes, personalization of treatment, improved disease monitoring and prognosis, and a potential for multi-dimensional diagnostics [31,32].”

Additionally, [outside the MS] investigating serum circulating miRNA in T2D patients is crucial due to the heterogeneous clinical presentation of the disease and the challenges associated with its treatment, which often relies on trial and error. T2D manifests differently in everyone, with varying degrees of insulin resistance, β-cell dysfunction, and associated comorbidities, making it difficult to adopt a one-size-fits-all approach for diagnosis and treatment. If specific circulating miRNAs can serve as biomarkers to subcategorize T2D patients into distinct subgroups, it would provide valuable insights into the underlying pathophysiology of the disease. This would enable a more personalized approach, allowing clinicians to tailor treatments based on the patient’s specific molecular profile, improving therapeutic efficacy and minimizing adverse effects.

  1. T2D subjects need to be better described for confounding factors, such as treatments and presence of T2D complications.

Response: As per reviewer’s recommendation, the T2D subject selection has been further detailed in the Materials and Methods section as follows.

Lines 356-363: “All patients included in the study tested negative for GAD antibodies (ELISA Test Kit; Demeditec Diagnostics, GmbH, Germany) and were diagnosed with T2D (fasting blood glucose (FBG) ≥125mgdL, Hemoglobin A1c (HbA1c) ≥6.5%) by a registered medical practitioner. Patients with conditions causing secondary diabetes and other types of diabetes (MODY, LADA) were excluded. The patients exhibited at least two co-morbidities (hypertension, hyperlipidemia, obesity) or complications (micro- and macrovascular) and were under treatment by different medications (metformin, thiazolidines, SGLT2 inhibitors, and GLP-1 agonists).”

A further description of treatments and complications has been provided in the Results as follows.

Lines 104- 114: “The clinical characteristics of the non-overlapping clusters are described in Table S1. In this sub-cohort of non-overlapping subgroups (n=151), 57% had hypertension, 58% had peripheral neuropathy, 35% had retinopathy, 14% had cardiovascular disease and 13% had chronic kidney disease. Most T2D patients tested were receiving multiple hypo-glycemic drugs. The frequency at which these drugs were used in the sub-cohort of non-overlapping subgroups (n=151) were as follows: Metformin (89%), SGLT2 Inhibitors (64%), DPP-4 Inhibitors (44%), Glipizide (40%), Insulin (38%), GLP1-Receptor Agonists (30%), and Thiazolidiedione (15%). A minimum of 66% of patients were receiving 3-4 drugs simultaneously. In addition, 87% were receiving lipid lowering drugs and 56% were receiving antihypertensive treatment.”

  1. Serum is not optimal for microRNA quantification, since it contains all the molecules released upon platelet activation. Are authors able to quantify “plasma” circulating microRNAs in at least a subset of samples, to make sure they can obtain similar data? Furthermore, the haemolysis issue is also crucial here, since miR-451 is among the differentially expressed miRNAs, but it is also known to be highly expressed in haemolyzed plasma/serum samples.

Response: We have chosen serum as a medium for our study since serum has been used successfully in several miRNA biomarker studies (References 29, 30 in the manuscript).

Lines 76-79: “When compared with human plasma, serum, which is often described as a full-body biopsy contains higher miRNA concentrations [29] and higher stability of circulating extracellular miRNAs even after multiple freeze-thaw cycles when archived for long-term storage [30].”

The plasma samples of the study cohort are not available to us. Hence, we are not able to quantify “plasma” circulating miRNAs in a subset of “serum’ samples to make sure we obtain similar data, as suggested by the Reviewer. However, we assure the Reviewer that the serum samples were collected by professional phlebotomists in plain tubes without anticoagulants. Following centrifugation, samples were thoroughly checked for evidence of hemolysis, visually. Any sample showing hemolysis (non-yellow shade) was discarded and a replacement was requested immediately.

Although miR-451 can be expressed in hemolyzed samples due to red blood cells rupture, studies have shown that it can be upregulated in patients with T2D and non-alcoholic fatty liver disease (Lines 288- 290).

Lines 288- 290: “The upregulation of miRNA-451a in T2D patients was consistent with prior studies which showed that it was also elevated in patients with non-alcoholic fatty liver disease, increasing their risk for T2D [55,56].”

In our study, miR-451a was upregulated in only the SIRD subgroup and therefore does not seem to be a contaminant.

  1. Patients’ description. Table 1 is not clear enough: explain the classification based on the unsupervised clustering approach. The reference 20 is too relevant for this present work for just being cited. Please explain the bases for your present effort.

Response: Reviewer’s suggestion accepted, and details of Reference 20 (Bayoumi et al., 2024) have been provided in the manuscript as follows.

Lines 94- 123: “The cohort characteristics of T2D patients of this study have been reported else-where [20]. Briefly, patients in the total T2D cohort tested (n=348) had a mean duration of T2D of 14 years and had multiple comorbidities and complications; with every patient exhibiting at least 2 complications. Using unsupervised soft cluster analysis by Auto Cluster IBM Modeler in the SPSS software, the 348 Emirati T2D patients were subtyped by Bayoumi et al. [20] by employing five etiological predictor variables: FBG, FSI, BMI, HbA1c and age at diagnosis. Of the five clusters identified, the first four matched Ahlqvist et al [14] subgroups (SIRD, SIDD, MARD, and MOD). A fifth new subtype MEOD was identified. An extensive overlap between these five clusters was observed. While 151/348 patients (43%) appeared in five distinct clusters, the remaining 197/348 patients (57%) showed varying degrees of overlap, with individuals appearing in two or more clusters. The clinical characteristics of the non-overlapping clusters are described in Table S1. In this sub-cohort of non-overlapping subgroups (n=151), 57% had hyper-tension, 58% had peripheral neuropathy, 35% had retinopathy, 14% had cardiovascular disease and 13% had chronic kidney disease. Most T2D patients tested were receiving multiple hypoglycemic drugs. The frequency at which these drugs were used in the sub-cohort of non-overlapping subgroups (n=151) were as follows: Metformin (89%), SGLT2 Inhibitors (64%), DPP-4 Inhibitors (44%), Glipizide (40%), Insulin (38%), GLP1-Receptor Agonists (30%), and Thiazolidiedione (15%). A minimum of 66% of patients were receiving 3-4 drugs simultaneously. In addition, 87% were receiving lipid lowering drugs and 56% were receiving antihypertensive treatment.

For the purpose of this study, nine patients at the peak of the distribution of each of the five non-overlapping phenotypic clusters (45/151) were selected. The primary clinical criterion identifying each subgroup is confirmed in the 45 selected patients (Table 1). The median age of diagnosis of T2D was oldest in the MARD subgroup (65 years) and youngest in MEOD (34 years). MOD cluster had a mean BMI of 41.5 Kg/m2. SIDD showed the highest mean fasting blood glucose of 281 mgdL, a mean HbA1c of 10.4% and the lowest mean β-cell function (HOMA-B) of 15. SIRD had the highest mean insulin resistance (HOMA-IR) of 15.1. All 45 T2D patients selected had two or more complications and undergoing treatment with 3-4 oral hypoglycaemic drugs simultaneously.”

The following supplementary table (Table S1) has been added to the Supplementary materials.

Supplementary Table S1: Clinical characteristics of 151 T2D patients in the non-overlapping apices of distribution of clusters, identified by unsupervised clustering approach in the cohort of 348 patients with long-standing Type 2 diabetes (Bayoumi et al., 2024).

Cluster 1

Cluster 2

Cluster 3

Cluster 4

Cluster 5

Severe Insulin Resistant Diabetes [SIRD]

Severe Insulin Deficient Diabetes [SIDD]

Mild Age-Related Diabetes [MARD]

Mild Obesity-Related Diabetes [MOD]

Mild Early Onset Diabetes [MEOD]

Number of patients (n)

25

40

24

23

39

Mean (±SD)

Mean (±SD)

Mean (±SD)

Mean (±SD)

Mean (±SD)

Age at diagnosis (years)

45.3 (11.7)

36.7 (8.6)

58.33 (7.4)

35.2 (8.7)

32.8 (5.9)

BMI (Kg/m2)

33.9 (3.9)

32.1 (6.7)

28.3 (3.6)

38.7 (5.9)

26.8 (2.7)

HOMA-IR

16.3 (6.5)

8.1 (5.0)

3.0 (1.7)

3.6 (2.6)

2.3 (1.5)

HOMA-B (%)

254 (267)

32 (19)

111 (157)

118 (111)

69 (43)

Fasting Blood Glucose (mgdL)

151 (38)

238 (69)

114 (21)

115 (30)

113 (23)

Fasting Insulin (mIU/mL)

43.1 (9.6)

13.5 (6.7)

10.3 (4.2)

12.1 (6.7)

8.3 (4.6)

HbA1c (%)

7.3 (0.9)

10.7 (2.3)

6.3 (0.7)

6.9 (0.8)

6.6 (0.9)

  1. The sentence “On applying a read count cut-off of ≥10 per sample the total number of identified miRNAs reduced from 1182 to 430” is unclear. Please explain better. It would be relevant to evaluate how many miRNAs are co-expressed in all samples.

Response: Reviewer’s suggestion accepted and modified for better clarity as follows.

Lines 142- 146: “A total of 1182 miRNAs were detected in the 50 samples. The miRNAs that had ≥10 number of copies (n= 430) were selected for downstream analysis. Of these, 253 miRNAs (58.83%) were expressed in all samples, 16 miRNAs in only controls and the remaining 161 in either one or multiple T2D clusters (Figure S1).”

  1. The legend of the heatmap of Figure 3 is misleading. It is not clear if T2D and Controls are subdivided in vertical (by colour) or in horizontal (by names on the side). Please clarify.

Response: Reviewer’s comment accepted. The T2D and control grouping has been clearly defined in the re-labelled Figure 2 legend (Lines: 163- 169).

The double labels have been removed from the figure to avoid confusion.

Figure 2. Differential expression profile of circulating miRNAs in Type 2 Diabetes versus Control. The log2Fold change (log2FC) of each differentially expressed miRNA between the Control and Type 2 diabetes (T2D) groups is shown as heatmap where the red indicates upregulated miRNAs, blue represents downregulated miRNAs and white represents no expression change. The two vertical groups correspond to Control (Green bar) and T2D (Yellow bar). The individual vertical column in each group indicates the results of a single patient or subject and each horizontal row represents a single miRNA.

  1. A validation phase (also by a different platform of quantification, such as RT-qPCR) may significantly strengthen the study. Moreover, authors may give a hint on the actual number of miRNA molecule per serum volume unit. This is relevant, since a biomarker should also be robustly expressed and easily detectable.

Response: As suggested by the reviewer, we understand that a validation of the miRNA expression by RT-PCR can strengthen the study. Unfortunately, this study was conducted in only a subset of samples, and we are not able to perform the RT-PCR at the moment. This limitation of the current study has been reported in the discussion in lines 328-334. However, we will consider this suggestion of expression validation by RT-PCR and quantitative measurement of miRNAs per serum volume unit for our future work in a larger cohort of samples.

Lines 328-334: “However, this study has its own limitations. Although hemolysis of serum samples was negated visually, a more sophisticated method like spectrometry can ensure no hemolysis. We also cannot rule out the potential release of platelet derived miRNAs in the serum samples since RNA released during the coagulation process may change the true repertoire of circulating miRNA [29]. The results are restricted to a small subset of samples that represent each T2D subtype warranting a validation of the expression patterns in a larger sample size by RT-qPCR.”

  1. Discussion is vague. Which of the miRNAs here identified are also found dysregulated (and with the same direction) in other studies?

Response: Reviewer’s suggestion accepted, and Discussion Section modified for better clarity as follows. It has been framed to discuss the results in a T2D cluster-wise manner.

Similar results are reported as:

Lines 288- 292: “The upregulation of miRNA-451a in T2D patients was consistent with prior studies which showed that it was also elevated in patients with non-alcoholic fatty liver disease, increasing their risk for T2D [55,56]. Moreover, hsa-miR-30e-3p, downregulated in the SIRD subtype, has been consistently downregulated in several human profiling studies and linked to prediabetes [24, 57].”

Lines 298- 299: “Consistent with our result, He et al. [60] have reported hsa-miR-20b-5p to be downregulated in T2D.”

Lines 304- 306: “In previous studies, hsa-miR-378c has been shown to be downregulated in T2D [60], regulate adiponectin expression, and play a role in T1D pathogenesis [63,64].”

Lines 307- 309: “Previous research has shown that hsa-miR-320a-3p which was upregulated in MOD cluster is also elevated in T2D patients and negatively correlates with BMI, FBG, and HOMA-IR in obese individuals [65].”

Lines 312- 317: “The MEOD cluster had the highest number (n= 19) of differentially expressed miRNAs, with only one down-regulated miRNA (hsa-miR-486-5p) known to promote pancreatic cell proliferation, increase insulin sensitivity, and inhibit apoptosis [67] and which was also reported as downregulated in T2D by He et al. [60]. The has-miR-144-5p upregulated in MEOD has also been shown to be upregulated in T2D patients and inhibit the expression of insulin receptor substrate 1 [50].”

Contrasting results are reported as:

Lines 277- 280: “While some miRNAs like hsa-miR-937, hsa-miR-615-3p, and hsa-miR-548ag were downregulated in this study, they have been reported as upregulated in others, poten-tially contributing to metabolic disorders, ischemic heart disease, and obesity-associated T2D [44,45].”

Lines 292-293: “In contrast to our findings, miR-221 has been reported as upregulated in both T2D and metabolic syndrome patients [58,59].”

Additional Changes:

In addition to the reviewers’ comments, we have made additional changes to improve the manuscript.

  1. Addition of Supplementary Table S1.
  2. Shifting of Figure 2 to Supplementary Figure S1 and subsequent re-numbering of Figures in the main manuscript and Supplementary figures.
  3. Replaced Figure 2 and Figure 5 with higher quality figures.
  4. Additional corrections in the main manuscript text as highlighted in yellow.
  5. Removal of duplicated and mis-labelled references.
  6. Rearrangement of references in ascending order as they appear in the main manuscript.

We believe these revisions address the reviewers' concerns and improve the manuscript's overall contribution to the field. We have attached a revised version of the manuscript with changes highlighted for your convenience.